# AQP5-1364A/C Polymorphism Affects *AQP5* Promoter Methylation

**DOI:** 10.3390/ijms231911813

**Published:** 2022-10-05

**Authors:** Katharina Rump, Theresa Spellenberg, Alexander von Busch, Alexander Wolf, Dominik Ziehe, Patrick Thon, Tim Rahmel, Michael Adamzik, Björn Koos, Matthias Unterberg

**Affiliations:** Klinik für Anästhesiologie, Intensivmedizin und Schmerztherapie, Universitätsklinikum Knappschaftskrankenhaus Bochum-Langendreer, 44892 Bochum, Germany

**Keywords:** AQP5, AQP5-1364 A/C polymorphism, sepsis, epigenetic, methylation, *AQP5* promoter, rs3759129

## Abstract

The quantity of aquaporin 5 protein in neutrophil granulocytes is associated with human sepsis-survival. The C-allele of the aquaporin (*AQP5*)-1364A/C polymorphism was shown to be associated with decreased AQP5 expression, which was shown to be relevant in this context leading towards improved outcomes in sepsis. To date, the underlying mechanism of the C-allele—leading to lower AQP5 expression—has been unknown. Knowing the detailed mechanism depicts a crucial step with a target to further interventions. Genotype-dependent regulation of AQP5 expression might be mediated by the epigenetic mechanism of promoter methylation and treatment with epigenetic-drugs could maybe provide benefit. Hence, we tested the hypothesis that *AQP5* promoter methylation differs between genotypes in specific types of immune cells.: *AQP5* promoter methylation was quantified in cells of septic patients and controls by methylation-specific polymerase chain reaction and quantified by a standard curve. In cell-line models, AQP5 expression was analyzed after demethylation to determine the impact of promoter methylation on AQP5 expression. C-allele of *AQP5*-1364 A/C promoter polymorphism is associated with a five-fold increased promoter methylation in neutrophils (*p* = 0.0055) and a four-fold increase in monocytes (*p* = 0.0005) and lymphocytes (*p* = 0.0184) in septic patients and healthy controls as well. In addition, a decreased *AQP5* promoter methylation was accompanied by an increased AQP5 expression in HL-60 (*p* = 0.0102) and REH cells (*p* = 0.0102). The C-allele which is associated with lower gene expression in sepsis is accompanied by a higher methylation level of the *AQP5* promoter. Hence, *AQP5* promoter methylation could depict a key mechanism in genotype-dependent expression.

## 1. Introduction

Sufficient immunity is a basis for coping with infectious diseases. However, an excessive immune response to an inflammatory stimulus might lead to additional harm as expected in sepsis [1] or acute respiratory distress syndrome [2]. Such a serious medical state is characterized by high mortality, despite intense research and medical treatment efforts [3]. Moreover, a wide variability exists regarding the outcomes, and this may be co-induced by genetic diversity [4,5]. A potential candidate gene for the investigation of genetic differences is the gene encoding aquaporin (AQP) 5 [4], which mediates key mechanisms of inflammation that prevail in sepsis and acute respiratory distress syndrome, including cell migration and proliferation [6], activity of the renin-angiotensin-aldosterone system [7], and the transmembrane transport of water [8]. We described a functional single nucleotide (-1364A/C, rs3759129) polymorphism in the *AQP5* gene promoter previously [7]. The C-allele (CC and AC) of this polymorphism occurs in around 28% of all individuals (Hardy-Weinberg-equilibrium) [9]. The substitution of C for A at position-1364 was associated with decreased *AQP5* mRNA and AQP5 protein expression and decreased neutrophil migration [10] which we ascribe to different water-influx followed by aberrant formation of lamellipodia. We expect that the impact on immune cell migration is the key for being an independent prognostic factor for decreased 30-day mortality in patients with sepsis with a hazard ratio of nearly 4 for the AC/CC genotypes compared with the homozygous AA genotype [4]. Accordingly, these results suggest a protective impact of a lower *AQP5* expression in sepsis [10], and mechanisms linked to an altered *AQP5* expression are of great interest. However, the underlying molecular and pathophysiological alterations linking a decreased AQP5 expression to the *AQP5* (-1364) genotype are still unknown. One possible regulatory mechanism might be a genotype-dependent DNA methylation of the cytosine residue in 5′-cytosine-phosphate-guanine-3′ (CpG) sites, which depicts a frequent epigenetic modification involved in the regulation of gene expression [11] and also influences *AQP5* expression [12,13]. We recently described a functionally important *AQP5* promoter methylation site (nt-937) linked to the binding of the inflammatorily acting nuclear transcription factor NF-κB, accompanied by increased methylation in sepsis non-survivors [14]. This demonstrates the potential role of epigenetic changes in *AQP5* promoter regions. Furthermore, the higher methylation of the *AQP5* promoter diminished reporter gene transcription, whereas demethylation by 5-azacytidine evoked higher *AQP5* expression [13]. Consequently, the *AQP5* promoter methylation might be influenced by the *AQP5* (-1364A/C) genotype and could be a mechanism influencing *AQP5* expression in sepsis. Accordingly, we tested the hypotheses that DNA methylation of the *AQP5* promoter is associated with the *AQP5* (-1364A/C) genotype. Keeping in mind that the effect of altered migration could be shown in neutrophils, we examined whether this regulation is restricted to special types of immune cells—a question that has, to the best of our knowledge, never been analyzed.

## 2. Results

The baseline characteristics of sepsis patients on the day of ICU admission are shown in Table 1. Seven patients with AA genotype and three patients with AC genotype were enrolled with homogeneous characteristics, including gender and age. All patients were white Germans of Caucasian ethnicity. Patients did not differ in disease-related parameters, such as C-reactive protein (*p* = 0.08988), procalcitonin (*p* = 0.5719) or Sequential Organ Failure Assessment score (*p* = 0.1398). It was noteworthy that the leucocyte count was higher in AA genotype carriers (*p* = 0.0414). The healthy control did not differ from septic patients in genotype and gender, but in age (Table 2). 

### 2.1. Cell Type-Specific Methylation of the AQP5 Promoter

In a first step, we analyzed the *AQP5* promoter methylation in different immune cells. Lymphocytes showed the highest promoter methylation compared to monocytes and neutrophils in healthy controls and septic patients. The methylation of lymphocytes was nearly sixfold higher compared to neutrophils (*p* = 0.0073, Figure 1a) and more than four times higher than in monocytes (*p* = 0.154, Figure 1a) in healthy controls. A similar effect could be detected in cells of septic patients, where lymphocytes also showed higher methylation compared to neutrophils (*p* = 0.0055, Figure 1b) and monocytes (*p* = 0.0262, Figure 1b).

### 2.2. AQP5-1364A/C Genotype-Dependent AQP5 Promoter Methylation

In a second step, we analyzed the *AQP5* promoter methylation depending on the *AQP5-1364A/C* promoter polymorphism. The *AQP5* promoter methylation was higher in healthy controls and septic patients carrying C-allele compared to A-allele within all cell types analyzed (Figure 2). The methylation of healthy C-allele carriers was more than threefold higher in neutrophils (*p* = 0.0013; Figure 2a) and monocytes (*p* = 0.0020; Figure 2a) and more than twofold higher in lymphocytes (*p* = 0.0182; Figure 2a). This association also remained stable in septic patients. Neutrophils of C-allele carriers showed more than fourfold increased methylation compared to A-allele carriers (*p* = 0.0055, Figure 2b), while the increase in monocytes was nearly fourfold (*p* = 0.0005; Figure 2b) and threefold in lymphocytes (*p* = 0.0184; Figure 2b). This effect remains stable when combining healthy-and sepsis-probands (neutrophils AA vs. AC *p* =0.002; monocytes AA vs. AC *p* < 0.001 and lymphocytes AA vs. AC *p* = 0.02). Regarding healthy individuals compared to sepsis patients we find a tendency towards higher methylation levels in sepsis in all cell-types (neutrophils sepsis (0.067) vs. neutrophils healthy (0.035) *p* = 0.268, monocytes sepsis (0.090) vs. monocytes healthy (0.059) *p* =0.345, and lymphocytes sepsis (0.3954) lymphocytes healthy (0.129) *p* = 0.035.

### 2.3. Expression and AQP5 Promoter Methylation after ADC Incubation

In order to elucidate if AQP5 expression is indeed regulated by methylation, we incubated two proliferating lymphatic cell lines HL-60 and REH with the DNA-demethylating agent ADC(5-Azacytidine). The ADC resulted in a 10% decreased methylation (*p* = 0.0309; Figure 3a) and an increased *AQP5* mRNA expression (*p* = 0.0102, Figure 3b), also leading towards an increased AQP5 protein expression (Figure 3c) in HL-60 cells. A similar effect could be detected in REH-cells, where a 10% decreased methylation (*p* = 0.0004; Figure 3d) doubled *AQP5* mRNA (*p* = 0.0460; Figure 3e) and protein expression (Figure 3f).

## 3. Discussion

With this study we could show, that the CpG methylation of the *AQP5* promoter correlates strongly to the *AQP5* A/C-1364 genotype. As we also showed that AQP5 expression is related to *AQP5* promoter methylation, this could depict the mechanistic explanation for the decreased AQP5 expression in C-allele carriers, which we showed previously [7,10]. In addition, this study indicates a differential *AQP5* promoter methylation in different types of immune cells. While we expected to find an altered methylation under sepsis-conditions, this could only be detected as a tendency missing the level of significance in the analyzed cells types despite the lymphocytes. In our study, healthy individuals showed lower methylation but this might also be ascribed to be a possible effect of lower age in this group.

In the past, we could not elucidate the mechanism that contributes to differential AQP5 expression in A- and C-allele carriers of the *AQP5*-1364A/C polymorphism. The transcription factor NMP4 seemed to bind to both alleles with comparable efficacy, and reporter assays in HEK293 cells showed no significant differential allelic activity [16]. Here, we demonstrate, an explanatory approach linking the polymorphism to *AQP5* promoter methylation in the promoter region nt-145 to nt-49, which is included within the highly active promoter region of the *AQP5* promoter from nt-52 to nt-634 [7].

We decided to analyze this part of the promoter, because of its CpG density and the previously described functional relevance [7]. The promoter area around position-1364 does not contain CpG island and was demonstrated to have low promotor activity in previous studies [7]. The correlation between a certain single nucleotide polymorphism and DNA methylation is not a newly described phenomenon. Several studies in the last few years have indicated that genetics can influence the epigenetic state [17]. It is also known that the DNA CpG methylation status is influenced by genetic polymorphisms in many genes [18]. As an example, single nucleotide polymorphism-associated methylation was identified in the promoter region of the TRPC3-gene [19]. It is assumed that a genotype that influences the local density of CpG dinucleotides might have the capability to alter the likelihood of nearby CpGs becoming methylated [19]. Changes in the promoter methylation in correlation with a single nucleotide polymorphism could influence the expression of its target gene by affecting the binding of certain proteins to the DNA sequence [20].

Within our previous studies we showed in that the C-allele of the *AQP5* promoter is associated with decreased AQP5 expression [7,10]. We now demonstrate a mechanistic link regarding the way in which the AQP5 expression in immune cells might be regulated. We confirm previous results that demonstrate CpG methylation in the *AQP5* promoter region to regulate the *AQP5* gene expression [13]. Nomura et al. showed that murine cells in a hypomethylated state express high levels of AQP5, while highly methylated cells are associated with a decreased AQP5 expression [13]. As a mechanistic link, they revealed an increased binding of the transcription factor SP1 in cultured mouse cells in the hypomethylated state [21].

Additionally, we show that *AQP5* promoter methylation differs in immune cells. This could depict one explanation for the differential expression of AQP5 in lymphocytes [22]. Our studies indicate a high AQP5 expression in the B-lymphoblastic cell line REH and a lower expression in the neutrophil-like cell line HL-60 and monocytic THP-1 cells [10,23]. The AQP5 expression seems to be induced during lymphocyte activation [22]. In addition, our and other studies indicate that AQP5 expression in dendritic cells, which can be derived from monocytes, and neutrophils may play a role in the cellular volume regulation [22]. Moreover, the presence of AQP5 in A549 cells, an alveolar epithelial cell line, is associated with increased neutrophil migration [24].

Differential expression of AQP5 due to the methylation density might impact on immune cell migration, a possible key mechanism in sepsis [25]. Aquaporins facilitate cell migration by mediating water influx into membrane protrusions, which causes actin reorganization and the formation of lamellipodia, which provides a foundation for the cell to move forward [26]. Neutrophil cells and macrophages, derived from monocytes, are most prominent to migrate into tissue via lamellipodia formation [27,28,29], whereas this mechanism is not predominantly described in lymphocytes [30]. As a former study demonstrated the role of promoter methylation in inflammatory cells in lung-disease [31]. Our recent study indicates low methylation in neutrophils and monocytes, which could cause a high expression in these cells. In turn, lymphocytes seem to have a higher *AQP5* promoter methylation. Furthermore, in our study, *AQP5* promoter methylation seemed to be similar in cells from healthy controls and septic patients. It is of note that the downregulation of AQP5 expression in septic patients might be an adaptive reaction of the immune system that could dampen inflammation-induced harm in diseases with an overwhelming inflammation or a dysregulated immune response [6]. This is in line with our previous results that show intense inflammation and a worsened outcome in A-allele-carriers during severe inflammation (sepsis) [4].

Regarding the clinical application of our findings, we can only speculate. As the A-allele is associated with decreased promoter methylation, increased AQP5 expression [10] and worse outcome in sepsis, the treatment with epigenetic drugs could maybe provide benefit [32]. Here, a targeted methylation of A-allele carriers might be beneficial in sepsis treatment. In addition, another interesting aspect which has to be considered is, if the C-allele may be less prone to develop sepsis, as we see the genotype dependent differences in methylation in healthy subjects and sepsis patients.

Some limitations of our study should be considered. Firstly, tumor cell lines were chosen for demethylation experiments. This is only limited transferable, while primary cells from healthy probands should have been utilized to translate the results to human pathology better. However, the incorporation of ADC in the DNA only works in proliferating cells and, hence, the usage of human primary blood cells would be not possible due to their nonproliferating properties. Therefore, results of our cell culture experiments may be limited in translation to human physiology. Furthermore, the differences in promoter methylation before and after ADC incubation and between the *AQP5*-1364A/C genotypes may appear small. Nevertheless, other studies have indicated that methylation rate differences below 10% can be decisive for the magnitude of gene expression [33,34]. In addition, the methylation seems to be higher in tumor cell lines than in primary cells. However, these values seem to be normal in leukemia or other cancer cells as the promoters of various genes in leukemia patients are hypermethylated [35]. Sepsis patients and healthy donors were significantly different in age. This might influence promoter methylation, but we found a similar methylation density in both groups in our study (*p* > 0.05).

## 4. Materials and Methods

### 4.1. Patients

After a positive vote by the Ethics Committee of the Medical Faculty of Bochum (Reg. No. 15-5457), 10 patients (age: 67.3 years ± 10.71 years [mean ± standard deviation], 7 men, 3 women) with the diagnosis of severe sepsis according to SEPSIS-2 definition [36] and 10 healthy control subjects (age: 29.5 years ± 9.4 years [mean ± standard deviation], 3 men, 7 women) were included in the study between 2014 and 2016. All ten sepsis patients were rechecked and hit the current SEPSIS-3 criteria [15] as well (suspected/proven infection and SOFA-score of two or more points). Written informed consent was obtained from all participating patients or their guardians and the healthy control probands, according to the Declaration of Helsinki, good clinical practice guidelines and local legislation requirements. Blood samples were collected within 24 h of sepsis diagnosis [37].

All patients were followed up for 30-day survival. Clinical and demographic data, including the Simplified Acute Physiology Score II, and Sequential Organ Failure Assessment score were determined at the time of enrollment. All patients were treated in an ICU of a German university hospital according to recent guidelines and local standards.

### 4.2. Blood Sample Collection, Preparation and Storage

An amount of 50 mL whole blood (EDTA) was taken from patients after the fulfilment of the criteria of sepsis [15]. Blood samples were collected and processed immediately. Neutrophil granulocytes were separated from the whole blood by MACS (MACSxpress Neutrophil Isolation Kit, Miltenyi), and lymphocytes and monocytes after density gradient centrifugation (Ficoll Paque Premium 1.073, GE healthcare) with subsequent magnetic purification (Monocyte Isolation Kit, Miltenyi), according to the manufacturer’s instructions.

DNA was immediately extracted from the cells or from whole blood using the QIAamp^®^ DNA Blood Mini Kit (Qiagen, Hilden, Germany), respectively, according to manufacturer’s instructions. DNA samples were shock-frozen and stored at −80 °C until analysis. Separate sample aliquots were stored and thawed later for analysis of the current hypothesis and to avoid multiple freezing and thawing procedures.

### 4.3. Genotyping of Patients and Healthy Volunteers

Genotyping was performed as previously described with respect to *AQP5*(-1364) A/C polymorphism [7].

### 4.4. Overall DNA Methylation Analysis by Methylation-Specific Polymer Chain Reaction (PCR)

Methylation-specific PCR was performed to analyze the *AQP5* promoter methylation in the functionally important *AQP5* promoter region from nt-49 to nt-145, which had been analyzed in our group previously [14]. Accordingly, a Cells-to-CpG™ methylated and unmethylated gDNA control kit (Applied Biosystems, Foster City, CA, USA) was utilized to examine the primer specificity. Bisulfite conversion was performed using the EZ DNA Methylation-Gold™ Kit (Zymo Research). Gradient PCR was performed using gene-specific primers (Table 3) and the products were analyzed with agarose gel electrophoresis to determine the best PCR protocol settings.

Quantitative PCR with GoTaq^®^ qPCR Master Mix (Promega, Mannheim, Germany, cat. No. A6002) was used for the quantification of DNA methylation [38]. DNA dilution series containing methylated and unmethylated DNA was used to calculate a standard curve for the calculation of percentual by determining DeltaCt(Ct(U) − Ct(M)).

### 4.5. Methylation, mRNA and Protein Expression after 5-Aza-2′-Deoxycytidin (ADC) Incubation

Additionally, the methylation and expression of AQP5 in immortalized HL-60 and REH cell-lines after incubation with 50 μM ADC (Sigma-Aldrich, Taufkirchen, Germany) for 72 h was measured. Cells were lysed after ADC incubation. The DNA was extracted, and the methylation was quantified as described above. The RNA was extracted for expression analysis using the RNeasy MiniKit (Qiagen, Hilden, Germany), following the manufacturer’s instructions. An amount of 1 μg RNA was utilized to synthesize the cDNA with the QuantiTect Reverse Transcription kit (Qiagen, Hilden, Germany), according to the manufacturer’s instructions. Quantitative RT PCR was performed, as described above, using specific primers for *AQP5* and actin beta gene (reference) (Table 4). The data was analyzed using the delta-Ct method. Regarding protein quantification, the proteins were extracted by cell lysis with radioimmunoprecipitation buffer and shaking at 4 °C. After centrifugation, proteins were collected from supernatants and protein concentration was determined using a BCA Protein Assay (Thermo Fisher Scientific, Schwerte, Germany cat. no. 23225). An amount of 20 µg of protein was loaded per lane on a 10% sodium dodecyl sulfate gel, separated, transferred to a nitrocellulose membrane and an equal amount of protein was verified by Ponceau S staining. Western blot analysis was performed with antihuman AQP5 antibody (Santa Cruz Biotechnology, Dallas, TX, USA, cat. no. G-19, sc-9890), and an anti-actin antibody as a loading control, (clone C4, Millipore, Billerica Massachusetts, USA, cat. no. MAB1501R). Incubation with the first antibodies was performed simultaneously overnight. Li-Cor antibodies (anti-goat 800CW and anti-mouse 680CW) labeled with infrared dyes were used as secondary antibodies and imaging was performed using the Odyssey^®^ Imaging System (Li-Cor biosciences, Lincoln, NE, USA).

### 4.6. Statistics

The characteristics of patients are reported as numbers and percentages for categorical variables, means and standard deviations (±SD), or medians with interquartile ranges (25th; 75th percentile) for continuous variables, as appropriate. Categorical variables and continuous variables were compared by the chi-square or Fisher’s exact tests, Student’s *t*- or Wilcoxon Mann-Whitney tests, respectively, as appropriate. All variables assessed were tested for normal distribution using the Kolmogorov-Smirnov test.

All analyses were performed using SPSS (version 24, IBM, Armonk, NY, USA) and GraphPad Prism 8 (Graph-Pad, San Diego, CA, USA) was used for graphical presentations.

## 5. Conclusions

This study enabled us to elucidate a possible mechanism explaining the *AQP5* A/C-1364 promoter polymorphism impacts on AQP5 expression via altering *AQP5* promoter methylation. In addition, *AQP5* promoter methylation seem to depict a key mechanism in the AQP5 expression of immune cells. Lacking options to specifically influence single gene promoter methylation, this finding seems to offer an only limited therapeutic approach. But it was shown that an unspecific therapeutic intervention, such as 5-AZA treatment, in a mouse model might also be beneficial and reduce pulmonary edema in inflammatory acute lung injury [39]. This might also comprise the effect of reduced *AQP5* promoter methylation and increased expression. Without concluding a clear therapeutic approach, this, at least, offers first attempts that should be further studied. Our observational finding improves the insight into mechanisms underlying the regulation of inflammatory damage.

## Figures and Tables

**Figure 1 ijms-23-11813-f001:**
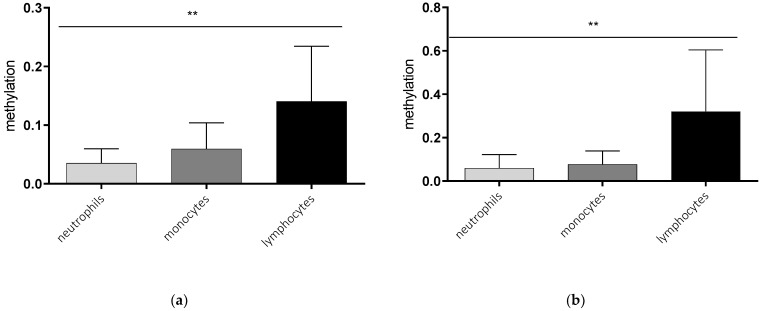
*AQP5* promoter methylation in the promoter region nt-145 to nt-49 in different blood cells of healthy controls (**a**) and septic patients (**b**). Neutrophils, monocytes, and lymphocytes were isolated using magnetic separation, DNA extraction, bisulfite conversion and methylation-specific PCR for the promoter region from nt-49 to nt-145. *AQP5* promoter methylation was significantly different between the immune cells analyzed in healthy controls ((**a**) *p* = 0.0039, *n* = 10) and septic patients ((**b**) *p* = 0.0041, *n* = 10), (** *p* ≤ 0.01).

**Figure 2 ijms-23-11813-f002:**
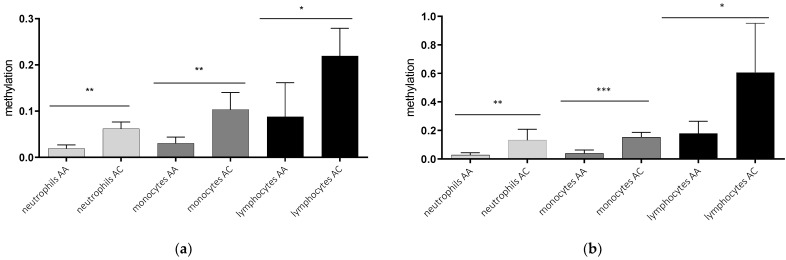
Genotype-dependent methylation in the promoter region nt-145 to nt-49 in different blood cells of healthy controls (**a**) and septic patients (**b**). Neutrophils, monocytes, and lymphocytes were isolated using magnetic separation, DNA extraction, bisulfite conversion and methylation-specific PCR for the promoter region from nt-49 to nt-145. Cells from C-allele carriers (*n* = 4) show higher methylation than cells from A-allele carriers (*n* = 3) in healthy controls (**a**) and cells from septic patients, where C-allele carriers (*n* = 3) showed higher methylation than A-allele carriers (*n* = 7) (* *p* ≤ 0.05; ** *p* ≤ 0.01; *** *p* ≤ 0.001).

**Figure 3 ijms-23-11813-f003:**
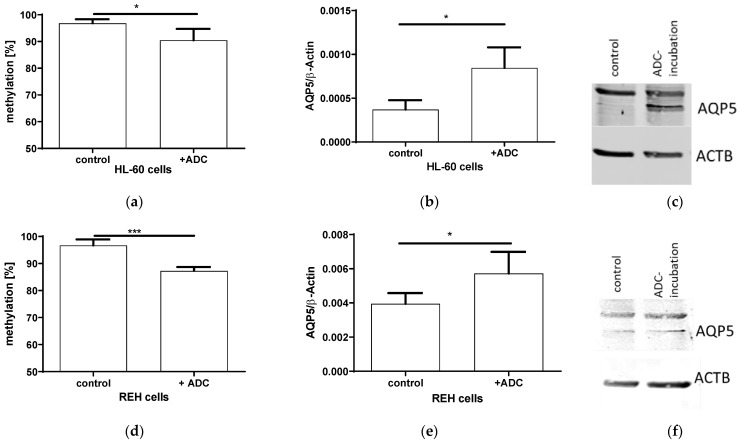
Incubation of proliferating immune cells HL-60 and REH with 50 µM aza-deoxycytosin (ADC) for 72 h. The *AQP5* promoter methylation in the promoter region nt-145 to nt-49 in HL-60 ((**a**) *n* = 4) and REH cells ((**d**) *n* = 4) decreased after ADC incubation. The *AQP5* mRNA expression increased in HL-60 ((**b**) *n* = 4) and REH cells ((**e**) *n* = 4). A similar effect could be detected in the protein expression in HL-60 (**c**) and REH (**f**) cells, where a representative bot out of three experiments is shown (* *p* ≤ 0.05; *** *p* ≤ 0.001).

**Table 1 ijms-23-11813-t001:** Baseline characteristics of septic patients and healthy controls.

**Septic Patients**	**AA Genotype (*n* = 7)**	**AC Genotype (*n* = 3)**	***p*-Value**
male/female	5/2	2/1	0.8803
age years (±SD)	59.1 (±21)	65.3 (±9.7)	0.6459
Hb (g/dL (±SD)) day 1	10.9 (±1.5)	10.1 (±2.3)	0.5228
leucocytes (×1000 µL (±SD))	14.06 (±6.7)	4.0 (±3.1)	0.0414
thrombocytes (×1000 µL (±SD))	209.6 (±136.3)	145.5 (±96.9)	0.4544
CRP (mg/dL (±SD))	13.2 (±12.6)	12.17 (±6.4)	0.8988
PCT (ng/L (±SD))	3.1 (±4.7)	1.4 (±1.9)	0.5719
temperature (°C (±SD))	37.7 (±1.5)	37.0 (±1.5)	0.5719
noradrenaline (mg/h (±SD))	0.95 (±0.8)	0.6 (±0.7)	0.5318
30-day survival *n* (%)	43%	33%	0.7782
survival on ITS *n* (%)	43%	33%	0.7782
Gram-positive pathogen (%)	28%	67%	0.2598
Gram-negative pathogen detection *n* (%)	14%	0%	0.4902
no pathogen	57%	33%	0.4902
SOFA score day 1	8.4 (1.59)	6.3 (2.49)	0.1398
**Healthy Controls**	**AA Genotype (*n* = 6)**	**AC Genotype (*n* = 4)**	***p*-Value**
male/female	(2/4)	(2/2)	0.999
age years ((±SD)	29.83 (±8.9)	28.75 (±5.6)	0.8365
qSOFA	0	0	

CRP: C-reactive protein; PCT: procalcitonin; SOFA: Sequential Organ Failure Assessment score; qSOFA: quick SOFA [15].

**Table 2 ijms-23-11813-t002:** Comparison between septic patients and healthy controls.

	Septic Patients	Healthy Controls	*p* Value
AA-genotype	7	6	0.6392
AC-genotype	3	4	
male	3	4	0.9999
female	7	6	
Alter (Jahre (±SD))	67.3 (±13)	29.5 (±9.4)	<0.0001

**Table 3 ijms-23-11813-t003:** Oligonucleotide pairs used for methylation-specific PCR.

Oligonucleotide Name	Sequence	Annealing Temperature
AQP5_M_SE	CGTTTTCGTCGTATTTATTTTTTTC	55°
AQP5_M_AS	CTCCTTCTTCATAATAACCGCGA	
AQP5_U_SE	TGTTTTTGTTGTATTTATTTTTTTT	55°
AQP5_U_AS	CACCTCCTTCTTCATAATAACCACAA	

**Table 4 ijms-23-11813-t004:** Oligonucleotide pairs used for quantitative PCR-*AQP5* mRNA analysis.

Oligonucleotide Name	Sequence	
AQP5 forward	5′-TCGGTTCAGCCCCGCTCACT-3′	60°
AQP5 reverse	5′-GCCACACGCTCACTCAGGCT-3′	
Actin forward	5′-CTGGAACGGTGAAGGTGACA-3′	60°
Actin reverse	5′-AAGGGACTTCCTGTAACAATGCA-3′	

## Data Availability

The data presented in this study are available on request from the corresponding author.

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
