# Peer review of "AQP5-1364A/C Polymorphism Affects AQP5 Promoter Methylation"

_ijms, 2022, doi:10.3390/ijms231911813_

Round 1

Reviewer 1 Report

Dear authors,

First of all, I’d like to give a great congratulation to them for nice and successful study. Rump et al. investigated whether AQP5 promoter methylation differs between geno-types in specific types of immune cells or not. They performed methylation-specific PCR of blood samples in sepsis patients and healthy control. Their trials are unique and original to overcome the unmet needs in the clinical practice and translational research.

However, there are several points to be corrected.

1.       The readers must be curious whether there should be specific genetic features having meaningful functions in practice rather than their methods. As it takes a long time to define the methylation of AQP5 gene promoter, it can be less informative to the physician who are treating sepsis patients than authors imagine. Authors had better explain the clinical application and future possibility of their findings.

2.       Abstract had better be corrected according to the routine formation. The abstract of this manuscript looks like an assay rather than scientific article. Please, reduce the volume of background, rewrite the methods and results more concretely.

3.       Readers may wonder what the definition of “severe sepsis” is. There was no explanation how to define and diagnosis the sepsis of patients.

Good luck.

Author Response

First of all, I’d like to give a great congratulation to them for nice and successful study. Rump et al. investigated whether AQP5 promoter methylation differs between genotypes in specific types of immune cells or not. They performed methylation-specific PCR of blood samples in sepsis patients and healthy control. Their trials are unique and original to overcome the unmet needs in the clinical practice and translational research.

Thank you very much for your kind and helpful comments. We considered all of them carefully and implemented, according to your proposals, the following within the manuscript.

  1. The readers must be curious whether there should be specific genetic features having meaningful functions in practice rather than their methods. As it takes a long time to define the methylation of AQP5 gene promoter, it can be less informative to the physician who are treating sepsis patients than authors imagine. Authors had better explain the clinical application and future possibility of their findings

  1. Within further work 1,2,3,4,5,6 we and others could show the prognostic impact of the AQP5 -1364A/C promoter-polymorphism. We could show its association to a reduced AQP-5 expression in C-Allele-carriers, showing less neutrophil-migration and worsened sepsis-survival. As you know, we investigated one possible mechanism, linking the polymorphism to the gene-expression level. This does even not exclude promoter methylation being maybe also influenced by additional mechanisms. We explain on this and on perspective therapeutic approaches one could imagine only in very brief words. As you propose, we now explain the potential meaning of promoter-methylation in more detail and we also add little more perspective, thinking about clinical applications.

So, we included in the abstract on line 20 ff: Genotype-dependent regulation of AQP5 expression might be mediated by the epigenetic mechanism of promoter methylation and treatment with demethylating drugs could maybe provide benefit.

In addition, we added a section in the discussion, where we speculated about the potential therapeutic implications in more detail on line 209 ff:

About the clinical application of our findings we can only speculate. As the A-allele is associated with decreased promoter methylation, increased AQP5 expression [9] and worse outcome in sepsis, the treatment with epigenetic drugs could maybe provide benefit [29]. Here a targeted methylation of A-allele carriers might be beneficial in sepsis treatment.    

  1. Abstract had better be corrected according to the routine formation. The abstract of this manuscript looks like an assay rather than scientific article. Please, reduce the volume of background, rewrite the methods and results more concretely.

  1. We reorganized the Abstract of the manuscript shifting the background information to the introduction of the text. We also improved the parts “methods” and “results” within the Abstract. In addition, we added p-values and more data to the abstract. (lines 13 ff.)

So, the whole abstract is now changed.

  1. Readers may wonder what the definition of “severe sepsis” is. There was no explanation how to define and diagnosis the sepsis of patients.

  1. Patients were recruited while sepsis-definition was revised and changed to the recent SEPSIS-3 definition. All included patients fulfilled the criteria of severe sepsis according to SEPSIS-2 and we verified that all of them hit the recent SEPSIS-3 definition. We clarified this within the text in lines 233 ff.

  1. Rahmel T, Rump K, Peters J, Adamzik M. Aquaporin 5 -1364A/C Promoter Polymorphism Is Associated with Pulmonary Inflammation and Survival in Acute Respiratory Distress Syndrome. Anesthesiology. 2019;130(3):404-413. doi:10.1097/ALN.0000000000002560
  2. Rump K, Adamzik M. Function of aquaporins in sepsis: a systematic review. Cell Biosci. 2018;8:10. doi:10.1186/s13578-018-0211-9
  3. Bergmann L, Nowak H, Siffert W, et al. Major Adverse Kidney Events Are Associated with the Aquaporin 5 -1364A/C Promoter Polymorphism in Sepsis: A Prospective Validation Study. Cells. 2020;9(4). doi:10.3390/cells9040904
  4. Adamzik M, Frey UH, Möhlenkamp S, et al. Aquaporin 5 gene promoter--1364A/C polymorphism associated with 30-day survival in severe sepsis. Anesthesiology. 2011;114(4):912-917. doi:10.1097/ALN.0b013e31820ca911
  5. Rump K, Unterberg M, Bergmann L, et al. AQP5-1364A/C polymorphism and the AQP5 expression influence sepsis survival and immune cell migration: a prospective laboratory and patient study. J Transl Med. 2016;14(1):321. doi:10.1186/s12967-016-1079-2
  6. Unterberg M, Rahmel T, Rump K, et al. The impact of the COVID-19 pandemic on non-COVID induced sepsis survival. BMC Anesthesiol. 2022;22(1):12. doi:10.1186/s12871-021-01547-8

Reviewer 2 Report

In the article “AQP5 -1364A/C Polymorphism affects AQP5 promoter methylation” by Rump et al authors describe a correlation between a -1364A/C polymorphism in the AQP5 gene and the DNA methylation levels of the AQP5 proximal promoter. In my opinion, although the work is promising the presented results seem rather preliminary. I have two main concerns:

1.       Size and composition of the studied cohort: The study is limited to 10 patients that are compared to 10 healthy individuals. The size of the cohort seems rather small. Moreover, I could not find a detailed description of the healthy donor’s genotype and their corresponding age, sex, and ethnicity. Does the statistical analysis performed by the authors correct for these potential differences? Authors should provide statistical proof that the results obtained using such a small cohort are meaningful.

2.       The study fails to show correlations between the levels of DNA methylation and the expression of AQP5 in the cell types isolated from patients and controls. What are the levels of AQP5 mRNA expression in these cell types? Do they correlate with the levels of DNA methylation? Regarding this point, the sentence “In addition, a decreased AQP5 promoter methylation was accompanied by an increased AQP5 expression in cell lines” in the abstract is misleading since this finding is not connected with the patient’s study.

In addition, there are interesting aspects in the results that the authors seem to disregard.  The healthy controls show lower methylation levels than the patients (although the authors claim is not significant). Does this correlate with lower levels of AQP5 expression in the patients? Is this gene regulated by sepsis? Moreover, the differences in methylation levels between genotypes appear obvious in the healthy donors. Might the AC genotype be more/less prone to developing sepsis?

Other minor concerns are:

There is important information missing in the introduction. For example, the frequency of the -1364A/C polymorphism. An introduction to the AQP5 promoter and regulation might be needed and it should be clearly explained that the authors are evaluating the methylation of the proximal promoter that likely contains a CpG island, not the -1364 area. Also, a reference in line 61 is missing.

Section 2.3 does not seem to add much to the research since it has been previously described that demethylating agents induce the expression of AQP5. Moreover, these experiments are giving rise to misleading sentences like “As we also showed that AQP5 expression is related to AQP5 promoter methylation” in line 135 and in the abstract, as mentioned above.

Did the authors perform any analysis to evaluate differential transcription factor binding to the -1364A/C polymorphism?

Line 102-103, “cell lines” should be “cell types”

Author Response

Reviewer 2

Thank you very much for your kind and helpful comments. We considered all of them carefully and implemented, according to your proposals, the following within the manuscript.

In the article “AQP5 -1364A/C Polymorphism affects AQP5 promoter methylation” by Rump et al authors describe a correlation between a -1364A/C polymorphism in the AQP5 gene and the DNA methylation levels of the AQP5 proximal promoter. In my opinion, although the work is promising the presented results seem rather preliminary. I have two main concerns:

  1. 1.Size and composition of the studied cohort: The study is limited to 10 patients that are compared to 10 healthy individuals. The size of the cohort seems rather small. Moreover, I could not find a detailed description of the healthy donor’s genotype and their corresponding age, sex, and ethnicity. Does the statistical analysis performed by the authors correct for these potential differences? Authors should provide statistical proof that the results obtained using such a small cohort are meaningful.

  1. During previous work on AQP5 1,2,3,4,5,6 we found a fundamental difference in gene expression in association to the described AQP -1364 A/C promoter-polymorphism. Until the current work we could not identify a possible mechanism, determining the difference in expression. Thus, we performed this work, labeled pilot-study and dealing with only low number of individuals. We detect a close and strong association between the polymorphism and promoter-methylation but we can only detect a quantitively difference between healthy patients and sepsis-conditions in the cell population of lymphocytes (p=0.0350). Anyway, we claim that promoter-methylation will be an important key towards lower AQP5 expression in C-Allele-carriers, being relevant during sepsis-circumstances.

As you propose, mainly information on the healthy individuals genotype is required. We added this information to our table 3 and as you see, maybe by chance, the frequency of C-Allele-carriers in our sepsis and healthy persons represents the natural distribution of frequency. We explained this within the introduction. In addition, the main result of our study is that the C-allele is associated with increased AQP5 promoter methylation. This effect was seen when considering both the healthy controls and septic patients individually. Due to your suggestion we added a new table comparing septic patients and healthy controls (see table 4). Our sepsis and control cohort do not differ in genotype (p= 0.6392) nor in gender (p=0.9999). However, it has to be considered that the age of the two cohorts differs (p<0.0001). Hence, we conducted a new analysis, where the data of septic patients and healthy controls were summarized in one analysis (n=20). As you see below the genotype dependent effect is even increased when combining these totally different cohorts.  These results were now included in the results section in lines 120 ff. They allow as to conclude that a) age has no influence on genotype dependent methylation and b) that already a cohort of 10 patients is capable to show meaningful differences in methylation.

  1. The study fails to show correlations between the levels of DNA methylation and the expression of AQP5 in the cell types isolated from patients and controls. What are the levels of AQP5 mRNA expression in these cell types? Do they correlate with the levels of DNA methylation? Regarding this point, the sentence “In addition, a decreased AQP5 promoter methylation was accompanied by an increased AQP5 expression in cell lines” in the abstract is misleading since this finding is not connected with the patient’s study.

In addition, there are interesting aspects in the results that the authors seem to disregard. The healthy controls show lower methylation levels than the patients (although the authors claim is not significant). Does this correlate with lower levels of AQP5 expression in the patients? Is this gene regulated by sepsis? Moreover, the differences in methylation levels between genotypes appear obvious in the healthy donors. Might the AC genotype be more/less prone to developing sepsis?

  1. Thank you for this essential comment. We did not want to confuse at this point. This study was set up as a pilot-study to measure DNA-methylation and genotypes in different blood cells of sepsis patients. The study protocol allowed us to collect 50 ml of blood of septic patients. The whole amount was needed to separate the three cell types with MACS technology and density gradient centrifugation. We used the whole cell fractions for DNA-isolation. Unfortunately, blood samples of septic patient show high interindividual differences in cell quality and quantity of the required cell types. Hence within an amount of 50 ml we were not able to split the three isolated cell types in analyses for DNA and RNA isolation. Hence, we did not measure m-RNA in patients’ samples. In the past we could already demonstrate in whole blood of septic patients that the C-allele which associated with an increased methylation in this study in all analyzed cell types is accompanied by a decreased expression in septic patients. This graph (see below) was not published here, as we already showed it elsewhere5. In addition, we agree that cell type specific AQP5 expression is an important issue to be measured in the future in primary cells.

However, to show an impact of methylation on AQP5 translation and expression, we choose a cell-culture model, because it offers the possibility of exterior influence of the methylation. This way we can demonstrate an effect of lower methylation on increasing AQP5 expression – but as you point out this is in a cell culture model. We explained this more detailed in the text, making sure that there is no further misleading information.

The tendency towards lower methylation in healthy persons comparing to sepsis was what we expected and what we were looking for. We now mention this tendency in the results and in the discussion section, which was only significant in lymphocytes (p=0.0350), so we would keep our way of interpretation. We show you the direct comparison here.

In the past we already examined AQp5 expression under septic stimuli. Here we demonstrated that the bacterial cell wall component fMLP increases AQP5 expression in the neutrophil cell line HL-605. In addition, we examined AQP5 expression in mice lungs and human monocytic cell line THP-1 after the inflammatory stimulus LPS and showed that this was decreased in a time-dependent manner. Hence, we conclude that this gene is regulated in the context of sepsis.

To your last question, we can only say as mentioned above that the C-allele shows better survival in sepsis but we cannot answer if it is less/more prone for sepsis. However, it seems an interesting point and we included this consideration in the text in the discussion section on line 220 ff.  

There is important information missing in the introduction. For example, the frequency of the -1364A/C polymorphism. An introduction to the AQP5 promoter and regulation might be needed and it should be clearly explained that the authors are evaluating the methylation of the proximal promoter that likely contains a CpG island, not the -1364 area. Also, a reference in line 61 is missing.

We added the natural underlying frequency of the C-Allele of AQP5 -1364 A/C in the introduction. The missing reference was included in line 61. We further explain which part of the promoter we considered and why we did so (from line 167).

Section 2.3 does not seem to add much to the research since it has been previously described that demethylating agents induce the expression of AQP5. Moreover, these experiments are giving rise to misleading sentences like “As we also showed that AQP5 expression is related to AQP5 promoter methylation” in line 135 and in the abstract, as mentioned above.

Again, thank you very much for your valuable comment. In section 2.3 we already explain that this experiment was performed using tumor cells. We know about the limited transferability, but showing that lower methylation leads to an increased expression of AQP5 still seems important to us, as this experiment shows us that we considered a regulatory region of the AQP5 promotor in our in vivo studies. We additionally stress the important information, in which experiment we measured the effect of demethylation on AQP5 expression and in which cells this was done.

Did the authors perform any analysis to evaluate differential transcription factor binding to the -1364A/C polymorphism?

Thank you again for this important question. Indeed, we studied transcription factor binding to the -1364A/C polymorphism with in silico analysis and reporter gene assay. We analyzed the effects of NMP47 (mentioned in the manuscript in lines 158 ff.), c-Myb and NFAT (data not published), which could not explain genotype dependent promoter activity. In previous experiments we could also show epigenetic influences on the binding of Nfk-B to the AQP5-promoter in the promoter region of nt-937, also leading to the question on further epigenetic regulation of AQP-5 expression being studied in this work, which is mentioned in the manuscript in lines 65 ff. 8.

Line 102-103, “cell lines” should be “cell types”

The incorrect designation in lines 102-102 (cell-lines vs cell types) was corrected.

  1. Rahmel T, Rump K, Peters J, Adamzik M. Aquaporin 5 -1364A/C Promoter Polymorphism Is Associated with Pulmonary Inflammation and Survival in Acute Respiratory Distress Syndrome. Anesthesiology. 2019;130(3):404-413. doi:10.1097/ALN.0000000000002560
  2. Rump K, Adamzik M. Function of aquaporins in sepsis: a systematic review. Cell Biosci. 2018;8:10. doi:10.1186/s13578-018-0211-9
  3. Bergmann L, Nowak H, Siffert W, et al. Major Adverse Kidney Events Are Associated with the Aquaporin 5 -1364A/C Promoter Polymorphism in Sepsis: A Prospective Validation Study. Cells. 2020;9(4). doi:10.3390/cells9040904
  4. Adamzik M, Frey UH, Möhlenkamp S, et al. Aquaporin 5 gene promoter--1364A/C polymorphism associated with 30-day survival in severe sepsis. Anesthesiology. 2011;114(4):912-917. doi:10.1097/ALN.0b013e31820ca911
  5. Rump K, Unterberg M, Bergmann L, et al. AQP5-1364A/C polymorphism and the AQP5 expression influence sepsis survival and immune cell migration: a prospective laboratory and patient study. J Transl Med. 2016;14(1):321. doi:10.1186/s12967-016-1079-2
  6. Unterberg M, Rahmel T, Rump K, et al. The impact of the COVID-19 pandemic on non-COVID induced sepsis survival. BMC Anesthesiol. 2022;22(1):12. doi:10.1186/s12871-021-01547-8
  7. Rump, K.; Siffert, W.; Peters, J.; Adamzik, M. The Transcription Factor NMP4 Binds to the AQP5 Promoter and Is a Novel Transcriptional Regulator of the AQP5 Gene. DNA Cell Biol 2016, 35, 322-327, doi:10.1089/dna.2015.3110.
  8. Rump, K.; Unterberg, M.; Dahlke, A.; Nowak, H.; Koos, B.; Bergmann, L.; Siffert, W.; Schafer, S.T.; Peters, J.; Adamzik, M.; et al. DNA methylation of a NF-kappaB binding site in the aquaporin 5 promoter impacts on mortality in sepsis. Sci Rep 2019, 9, 18511, doi:10.1038/s41598-019-55051-8.

Round 2

Reviewer 1 Report

Dear authors,

I’d like to give a great congratulation to them for nice and successful study. Authors corrected the weakness appropriately which was pointed out by reviewer. Then their study became well designed, and results got to be also reasonable and scientific, they can give enough helpful information to readers.

Good luck.

Reviewer 2 Report

The authors have addressed most of the concerns and have introduced the appropiate corrections.